Comparative analysis of deep learning algorithms for dental caries detection and prediction from radiographic images: a comprehensive umbrella review

Dashti Mahmood 1 dashti.mahmood72@gmail.com
Londono Jimmy 2
Ghasemi Shohreh 3
Zare Niusha 4
http://orcid.org/0000-0003-2256-8601 Samman Meyassara 5
Ashi Heba 5
Amirzade-Iranaq Mohammad Hosein 6
Khosraviani Farshad 7
http://orcid.org/0000-0003-0252-9584 Sabeti Mohammad 8
http://orcid.org/0000-0001-7998-7335 Khurshid Zohaib 9
1 Dentofacial Deformities Research Center, Research Institute of Dental Sciences, Shahid Beheshti University of Medical Sciences , Tehran , Iran
2 Department of Prosthodontics, Dental College of Georgia at Augusta University , Augusta, Georgia , United States
3 Department of Oral and Maxillofacial Surgery, Queen Mary College of Medicine and Dentistry , London , United Kingdom
4 Department of Oral and Maxillofacial Radiology, Islamic Azad University Tehran Dental Branch , Tehran , Iran
5 Department of Dental Public Health, College of Dentistry, King Abdulaziz University , Jeddah , Saudi Arabia
6 Faculty of Dentistry, Universal Scientific Education and Research Network (USERN), Tehran University of Medical Sciences , Tehran , Iran
7 UCLA School of Dentistry , Los Angeles, CA , United States
8 Department of Preventive and Restorative Dental Sciences, San Francisco School of Dentistry , San Francisco, CA , United States
9 Department of Prosthodontics and Dental Implantology, King Faisal University , Al Hofuf , Saudi Arabia
Raza Khalid
Electronic publication date: 2024 Nov 12
Publication date: 2024
Volume: 10
Electronic Location ID: e2371
Received 2024 Feb 5; Accepted 2024 Sep 9
Copyright: © 2024 Dashti et al.
Copyright year: 2024
Copyright holder: Dashti et al.
License: This is an open access article distributed under the terms of the Creative Commons Attribution License, which permits unrestricted use, distribution, reproduction and adaptation in any medium and for any purpose provided that it is properly attributed. For attribution, the original author(s), title, publication source (PeerJ Computer Science) and either DOI or URL of the article must be cited.
License URL: https://creativecommons.org/licenses/by/4.0/

Keywords: Dental caries, Dental radiograph, Diagnosis, Prediction, Artificial intelligence, Caries detection, Deep learning

Funding: The authors received no funding for this work.

==============================
Background

In recent years, artificial intelligence (AI) and deep learning (DL) have made a considerable impact in dentistry, specifically in advancing image processing algorithms for detecting caries from radiographical images. Despite this progress, there is still a lack of data on the effectiveness of these algorithms in accurately identifying caries. This study provides an overview aimed at evaluating and comparing reviews that focus on the detection of dental caries (DC) using DL algorithms from 2D radiographs.

Materials and Methods

This comprehensive umbrella review adhered to the “Reporting guideline for overviews of reviews of healthcare interventions” (PRIOR). Specific keywords were generated to assess the accuracy of AI and DL algorithms in detecting DC from radiographical images. To ensure the highest quality of research, thorough searches were performed on PubMed/Medline, Web of Science, Scopus, and Embase. Additionally, bias in the selected articles was rigorously assessed using the Joanna Briggs Institute (JBI) tool.

Results

In this umbrella review, seven systematic reviews (SRs) were assessed from a total of 77 studies included. Various DL algorithms were used across these studies, with conventional neural networks and other techniques being the predominant methods for detecting DC. The SRs included in the study examined 24 original articles that used 2D radiographical images for caries detection. Accuracy rates varied between 0.733 and 0.986 across datasets ranging in size from 15 to 2,500 images.

Conclusion

The advancement of DL algorithms in detecting and predicting DC through radiographic imaging is a significant breakthrough. These algorithms excel in extracting subtle features from radiographic images and applying machine learning techniques to achieve highly accurate predictions, often outperforming human experts. This advancement holds immense potential to transform diagnostic processes in dentistry, promising to considerably improve patient outcomes.

Introduction

Artificial intelligence (AI) is a field of science and engineering dedicated to developing intelligent machines that assist humans in performing complex tasks more efficiently and accurately (Toh, Dondelinger & Wang, 2019). This capability of AI to enhance productivity and accuracy spans various industries, optimizing operations and improving results (Toh, Dondelinger & Wang, 2019). Machine learning (ML), a branch of AI, enables systems to learn and evolve autonomously from experience, without explicit programming, by leveraging extensive datasets for algorithm training (Chen, Stanley & Att, 2020). This is achieved using large datasets to train algorithms, allowing them to continuously improve based on the information they analyze. ML systems analyze patterns and make decisions with minimal human intervention, adapting with increasing accuracy (Chen, Stanley & Att, 2020).

Deep learning (DL), a branch of ML, is built upon neural networks (NNs). These networks, modeled after biological neural networks, allow computers to recognize patterns in data (Schmidhuber, 2015). By layering these networks, DL models can learn from extensive datasets at multiple levels of complexity. This capability enables them to perform advanced tasks such as image and speech recognition with exceptional accuracy, mirroring certain aspects of human cognitive capabilities (Schmidhuber, 2015; Toh, Dondelinger & Wang, 2019).

A convolutional neural network (CNN) is a specific type of deep, feedforward network designed to process multi-array data, particularly images, through its layered architecture (Singh et al., 2023). This structured approach enables CNNs to excel in tasks involving the recognition and classification of visual inputs, rendering them essential in fields such as image and video recognition, medical image analysis, and other areas reliant on accurate pattern recognition from visual data (Singh et al., 2023).

In dentistry, AI has become crucial in oral diagnostics and treatment, considerably improving the accuracy and efficiency of these procedures. By conducting sophisticated analysis of radiographic images, AI facilitates accurate diagnosis and personalized treatment planning, which is essential in dental care. This integration of AI technology improves patient outcomes and optimizes workflow in dental practices, representing a transformative advancement in the management and delivery of dental health (Dashti, Ghasemi & Khurshid, 2023).

A frequent challenge in dental practices is the early detection of carious lesions and the initiation of preventive treatments to avoid more invasive procedures (Schwendicke, Tzschoppe & Paris, 2015). Dentists can use minimally invasive techniques to preserve tooth structure and maintain oral health by identifying decay in its earliest stages. This proactive approach prevents the progression of decay, reducing the need for more complex and expensive treatments later on. This underscores the importance of timely and effective intervention in dental care.

The International Caries Detection and Assessment System (ICDAS) defines dental caries (DC) as a “biofilm-mediated, non-communicable, multifactorial, diet-modulated, dynamic disease that results in a net mineral loss from dental hard tissues, influenced by biological, behavioral, environmental, and psychosocial factors, leading to the development of caries lesions” (Machiulskiene et al., 2020).

The detection of caries lesions commonly relies on visual and tactile examination techniques, often complemented by imaging methods such as radiography (the most commonly used adjunctive detection method), optical coherence tomography, intraoral scanning, quantitative light-induced fluorescence, or near-infrared light transillumination (Gomez, 2015; Michou et al., 2022).

While the adoption of new digital tools and methodologies in dentistry may require additional time and training for dentists and their staff (Khurshid, 2023), the application of AI, especially DL, can assist dental professionals in identifying and diagnosing caries lesions in images. AI offers self-guided learning, motivation, and immediate feedback, aiding users in assimilating knowledge effectively. This integration of AI in dental diagnostics not only improves the accuracy of caries detection but also enhances the educational experience for dental professionals, fostering deeper understanding and proficiency in managing oral health (Lee et al., 2018; Sarfaraz, Khurshid & Zafar, 2023). The field of DL in caries detection is highly dynamic, exhibiting substantial variations in methodologies and outcomes across studies. Several systematic reviews (SRs) and meta-analyses (MAs) have been performed to evaluate the accuracy, sensitivity, and specificity of DL algorithms in caries detection (Prados-Privado et al., 2020; Khanagar et al., 2021; Musri et al., 2021; Khanagar et al., 2022; Mohammad-Rahimi et al., 2022; Singh & Raza, 2022; Talpur et al., 2022). This umbrella review aims to assess and synthesize the conclusions drawn from these published SRs and MAs.

Survey methodology

Study design

This umbrella review uses SRs/MAs and carefully selected original studies (with specified inclusion criteria) to examine the accuracy, sensitivity, and specificity of various DL and CNN algorithms in detecting DC from radiographic images. The study methodology adheres to the “Reporting guideline for overviews of reviews of healthcare interventions” (PRIOR). Additionally, the preferred reporting items for systematic reviews and meta-analyses (PRISMA) flow diagram was used to illustrate the flow of study selection (Page et al., 2021; Gates et al., 2022).

Study type

This overview focuses specifically on SRs/MAs of original studies using various DL and CNN algorithms to predict DC from radiographic images. Table 1 provides a detailed summary of the research question addressed.

Table 1 Description of the PICO (P = Population, I = Intervention, C = Comparison, O = Outcome) elements.

Population	Radiographical images of patients who underwent examinations for DC.	
Intervention	Application of DL or CNN for DC detection in radiographic images.	
Comparison	Opinions of dental experts/specialists and established reference standards/models.	
Outcome	Assessment of DL or CNN performance in DC detection, measured by various accuracy estimates on radiographic images.	

Inclusion criteria

1) SRs comprising a minimum of two primary studies.

2) Focusing on the assessment of DL or CNN algorithm accuracy in detecting DC using radiographic images.

3) SR articles published up to August 2023.

Exclusion criteria

1) Conference proceedings, narrative review articles, and scientific posters available online.

2) SRs or MAs with ambiguous inclusion/exclusion criteria.

3) Studies presenting results outside the scope of this research.

4) SRs without a data extraction table or measurable outcome reporting.

Search strategy

The search process involved querying electronic databases such as Medline via PubMed, Scopus, Web of Science, and Embase for articles published up to the end of August 2023. Additionally, we reviewed articles published in languages other than English, translating them if necessary for inclusion. A comprehensive search syntax was developed using carefully selected keywords in different combinations with Boolean operators, and relevant MeSH terms were applied for PubMed. This syntax was adapted for other databases, following each database’s specific guidelines to ensure search accuracy and compatibility. Details regarding the number of results retrieved from each database are provided in Table 2.

Table 2 The search syntax used for each database and the number of results.

Database	Keyword	Result	
Pubmed	((“Dental Caries”[Mesh] OR “Dental Cavity” OR “Dental Decay” OR “Carious” OR “decay” OR “teeth caries”) AND (“Machine Learning”[Mesh] OR “Deep Learning”[Mesh] OR “Supervised Machine Learning”[Mesh] OR “Unsupervised Machine Learning”[Mesh] OR “Neural Networks, Computer”[Mesh] OR “artificial intelligence”)) AND (“Systematic Review” [Publication Type] OR “Systematic Reviews as Topic”[Mesh] OR “Systematic Review” OR “meta analysis” OR “meta-analysis”)	10	
Embase	(‘machine learning’/exp OR ‘machine learning’ OR ‘deep learning’/exp OR ‘deep learning’ OR ‘supervised machine learning’/exp OR ‘supervised machine learning’ OR ‘unsupervised machine learning’/exp OR ‘unsupervised machine learning’ OR ‘artificial neural network’/exp OR ‘artificial neural network’ OR 'artificial intelligence’) AND (‘dental caries’/exp OR ‘dental caries’ OR ‘dental cavity’ OR ‘dental decay’/exp OR ‘dental decay’ OR ‘carious’ OR ‘decay’ OR ‘teeth caries’) AND (‘systematic review (topic)’/exp OR ‘systematic review (topic)’ OR ‘systematic review’/exp OR 'systematic review’ OR ‘meta analysis (topic)’/exp OR ‘meta analysis (topic)’ OR ‘meta analysis’/exp OR ‘meta analysis’ OR ‘meta-analysis’/exp OR ‘meta-analysis’)	38	
Scopus	(TITLE-ABS-KEY (“Machine Learning”) OR TITLE-ABS-KEY (“Deep Learning”) OR TITLE-ABS-KEY (“Supervised Machine Learning”) OR TITLE-ABS-KEY (“Unsupervised Machine Learning”) OR TITLE-ABS-KEY (“Neural Network”) OR TITLE-ABS-KEY (“artificial intelligence”)) AND (TITLE-ABS-KEY (“Dental Caries”) OR TITLE-ABS-KEY (“Dental Cavity”) OR TITLE-ABS-KEY (“Dental Decay”) OR TITLE-ABS-KEY (“Carious”) OR TITLE-ABS-KEY (“decay”) OR TITLE-ABS-KEY (“teeth caries”)) AND (TITLE-ABS-KEY (“Systematic Review”) OR TITLE-ABS-KEY (“meta analysis”) OR TITLE-ABS-KEY (“meta-analysis”))	32	
Scopus secondary	(TITLE-ABS-KEY (“Machine Learning”) OR TITLE-ABS-KEY (“Deep Learning”) OR TITLE-ABS-KEY (“Supervised Machine Learning”) OR TITLE-ABS-KEY (“Unsupervised Machine Learning”) OR TITLE-ABS-KEY (“Neural Network”) OR TITLE-ABS-KEY (“artificial intelligence”)) AND (TITLE-ABS-KEY (“Dental Caries”) OR TITLE-ABS-KEY (“Dental Cavity”) OR TITLE-ABS-KEY (“Dental Decay”) OR TITLE-ABS-KEY (“Carious”) OR TITLE-ABS-KEY (“decay”) OR TITLE-ABS-KEY (“teeth caries”)) AND (TITLE-ABS-KEY (“Systematic Review”) OR TITLE-ABS-KEY (“meta analysis”) OR TITLE-ABS-KEY (“meta-analysis”))	6	
WOS	((TS = (“Machine Learning” OR “Deep Learning” OR “Supervised Machine Learning” OR “Unsupervised Machine Learning” OR “Neural Networks” OR “artificial intelligence”)) AND TS=(“Dental Caries” OR “Dental Cavity” OR “Dental Decay” OR “Carious” OR “decay” OR “teeth caries”)) AND TS=(“Systematic Review” OR “meta analysis” OR “meta-analysis”)	23	

Study selection and data extraction

Endnote X20 (Clarivate, Philadelphia, PA, USA) was used for citation management. After removing duplicate articles, two reviewers (H.A. and F.Kh.) performed the initial title and abstract screening phase. Any discrepancies were resolved through consultation with a third reviewer (J.L.). Subsequently, both reviewers independently evaluated the full texts of eligible studies, adhering to predetermined inclusion and exclusion criteria. Any disagreements during this phase were resolved through further discussions with the third reviewer. The PRISMA flow diagram depicting this process is displayed in Fig. 1.

Figure 1 PRISMA flow diagram of the selected studies.

Data extraction process

Two researchers (M.A. and Z.Kh.) were responsible for extracting data from the selected studies, while another pair (N.Z. and M.S.) verified all extracted data. Any disagreements were resolved through consensus, facilitated by a third reviewer (J.L.).

Table 3 summarizes the SRs and MAs included in this overview. The data extraction checklist includes the author’s name, year of publication, number of relevant articles included, databases searched, inclusion and exclusion criteria, and main outcomes.

Table 3 Summary of included SRs and MA.

Author/ year	Number of included relevant studies	Databases searched and time span	Inclusion criteria	Exclusion criteria	Main outcome	
Khanagar et al. (2021)	2	PM, MD, EM, CH, GS, SC, WOS, and SDL (January 2000 to March 15, 2020)	Articles should concentrate on artificial intelligence, specifically emphasizing its use in dental practices.

They should contain either predictive elements or quantifiable results for assessment purposes.

A clear description of the datasets employed in evaluating the model is essential.

	Articles focusing on topics outside the realm of artificial intelligence.

Manuscripts uploaded but not officially published.

Publications that only provide abstracts without the complete text.

Articles not composed in English.

	- AI tools offer substantial assistance to dental professionals, enhancing the quality of care delivered to patients.

- Dentists can leverage AI for improved diagnostic precision, treatment planning, and predicting patient treatment outcomes.

- General dentists can benefit from diagnostic guidance provided by deep learning systems.

- Automated processes can significantly save time and boost clinicians' efficiency.

- Utilizing these systems for secondary consultations can heighten diagnostic accuracy.

	
Khanagar et al. (2022)	18	PM, GS, SC, WOS, EM, CH, SDL (January 2000 until February 2022)	Original studies that focus on the use of AI-driven models to diagnose, detect, and predict dental caries.

Articles detailing the datasets utilized for these models’ training, validation, and testing.

Research with explicit details on measurable outcomes of the model’s performance.

Inclusion was not restricted by the study design type.

	Articles that provide only abstracts without the entire text.

Types of publications such as conference Articles, commentaries, editorials, brief communications, reviews, and scientific posters available online.

Research articles written in languages other than English.

	The effectiveness of these models in clinical settings for identifying patients at increased risk of dental caries (DC) is noteworthy. They assist in improving diagnostic and treatment processes as well as patient outcomes. The predictive models’ insights are valuable in devising preventive dental care and creating oral hygiene and diet plans for patients prone to DC. Despite their impressive performance, challenges exist due to data set size and diversity, as mentioned in many studies. Therefore, these models require further training and validation for optimal functioning.	
Mohammad-Rahimi et al. (2022)	5	PM (MD), GS, SC, EM, and ArXiv. (2010 up to 12th April 2021)	P: Research involving deep learning algorithms applied to dental images obtained from regular dental care, clinical trials, or studies on removed teeth.
I, C: Deep learning (neural network) based models for caries classification, detection, or segmentation, benchmarked against a standard test.
O: Any type of accuracy measurement at the level of the image, tooth, surface, or pixel.	Research lacking detailed information on the datasets used for model training and evaluation.

Studies that do not adequately describe the implemented deep learning framework.

Research that fails to distinguish the accuracy in identifying caries from detecting other dental conditions; this also includes review Articles.

	The reported accuracy of these studies is encouraging, though the quality of study design and reporting appears inadequate. Future research should focus on defining a standardized reference test and utilizing a broad, clinically relevant set of outcomes for comparison.	
Musri et al. (2021)	1	PM (MD), SL, and GS
(From 2015 to 2021)	Studies focusing on both permanent and deciduous teeth.

The utilization of a deep learning CNN with intraoral photographs.

Interpretation of radiographic images by experienced dental professionals.

Assessment of the effectiveness of deep learning CNNs in identifying dental caries.

Clinical research and scholarly articles.

Publications written in English.

	Research addressing dental conditions other than caries.

Use of artificial intelligence methods other than deep learning CNNs, and caries detection using panoramic, bitewing, or occlusal X-rays.

Review articles and systematic reviews.

	The reviewed studies often relied on limited data sets for both traditional and advanced deep learning analyses. Distinctions between early and root caries were not made clear. Accurate detection and diagnosis of dental caries are crucial for minimizing oral health management costs and increasing the chances of preserving natural teeth. This study’s findings indicate that deep learning CNN algorithms offer thorough, dependable, precise image analysis and disease identification. This enhances the diagnostic process, potentially improving dental caries prognosis in periapical radiographs.	
Prados-Privado et al. (2020)	4	PM (MD), IEEE Xplore, and SD. (up until 15 August 2020)	Comprehensive articles, encompassing conference reports, documenting the application of neural networks in caries detection and diagnosis.

Unrestricted by language or publication date.

	Literature reviews.

Studies not focusing on the application in dental caries.

Research lacking image data.

Studies not utilizing neural network technology.

	The diagnostic capabilities of AI models differ depending on the algorithm used, and their generalizability and dependability still require validation. For effective clinical implementation, it’s crucial to evaluate and compare the performance of each algorithm in various tasks.	
Singh & Raza (2022)	4	PM (MD) (From 2016 to December 30, 2020)	Complete versions of both research articles and conference proceedings.

Research employing neural network and deep learning techniques for interpreting dental and maxillofacial images, including tasks like tooth detection, tooth labeling/numbering, identifying dental caries, periodontal disease, endodontic treatments, and dental implants.

Inclusion limited to manuscripts published in English, with no constraints regarding the country of origin.

	Publications focusing on non-dental applications or not pertaining to dental and maxillofacial imaging.

Review articles.

Articles that are yet to be published (in press).

	AI and neural network applications in dental imagery have shown superior performance, sometimes even surpassing dental experts. Nonetheless, the accuracy of these systems needs to be confirmed across various imaging equipment and modalities to facilitate data standardization. Common dental conditions like tooth decay, periodontal issues, and apical lesions should be included in large datasets to enhance accuracy with minimal effort.	
Talpur et al. (2022)	4	PM, IEEE Xplore, SD, and GS databases. (Published between 2008 to 2022)	Articles must be written in the English language.

Research should focus exclusively on human dental health.

Studies need to detail the dataset sizes used in their research.

Articles must provide a detailed explanation of caries detection using machine learning techniques.

Studies should include information on various ML parameters, such as accuracy and classifier methods, used in identifying dental caries.

Eligible articles can include conference articles from IEEE.

	Articles written in languages other than English.

Studies focusing on the use of ML for dental X-ray image segmentation or general oral health identification.

Research linking DCs detection with any form of image processing tool.

Studies that are primarily about other dental conditions like oral cancer or tooth sensitivity.

Traditional methods like the ICDAS for detecting DC through visual examination of X-ray images.

Review articles, systematic reviews, meta-analyses, theses, dissertations, letters, editorials, abstracts, unpublished studies, case reports, small case series, and cross-sectional studies.

	From this systematic review, out of twelve studies, nine provided a high level of evidence in distinguishing between dental X-ray images with and without caries. The Neural Network Backpropagation Algorithm emerged as the most effective for dental image datasets, achieving up to 99% accuracy in caries detection. These results highlight the need for more comprehensive studies to diagnose various types of DC, categorized by progression (chronic, acute, arrested), indicating the severity and extent of the lesion. AI is poised to become a dominant technology for diagnosing a range of oral diseases, thanks to its ability to analyze extensive datasets with diverse records. AI models offer reliable information to dental professionals, enhancing the clinical decision-making process. The use of AI can lead to high-quality patient care and innovative dental research.	
Note:

PM, PubMed; MD, Medline; EM, Embase; CH, Cochrane; GS, Google Scholar; SC, Scopus; WOS, Web of Science; SDL, Saudi Digital Library; SL, SpringerLink; IEEE, Institute of Electrical and Electronics Engineers; SD, Science Direct.

Table 4 presents the data extracted from each included original article, encompassing details such as the author’s name, year of publication, image type, dataset size, model architecture, accuracy, sensitivity, specificity, precision, F1-score, positive predictive values (PPV), negative predictive values (NPV), and area under the ROC curve (AUC).

Table 4 Summary of specific included original articles.

Author, year	Image type	Dataset size	Architecture model	Accuracy	Sensitivity/ Recall	Specificity	Precision	F1-score/ Dice coefficient	AUC	ROC	PPV	NPV	
Devito, de Souza Barbosa & Filho (2008)	Bitewing	160	ANNs	Not reported	Not reported	Not reported	Not reported	Not reported	Not reported	0.884	Not reported	Not reported	
Lee et al. (2018)	Periapical	600	CNNs	0.82 (0.755–0.871)	0.81 (0.745–0.861)	0.83 (0.765–0.881)	Not reported	Not reported	0.845 (0.790–0.901)	Not reported	0.827 (0.761–0.879)	0.814 (0.750–0.864)	
Choi, Eun & Kim (2018)	Periapical	475	CNNs	Not reported	Not reported	Not reported	Not reported	0.74	Not reported	Not reported	Not reported	Not reported	
Cantu et al. (2020)	Bitewing	252	CNNs	0.80	0.75	0.83	Not reported	Not reported	Not reported	Not reported	Not reported	Not reported	
Geetha, Aprameya & Hinduja (2020)	Intraoral digital images	145	ANNs	0.971	Not reported	Not reported	0.987	Not reported	Not reported	Not reported	Not reported	Not reported	
Chen et al. (2021)	Periapical	2,900	CNNs	Not reported	Not reported	Not reported	Not reported	Not reported	Not reported	Not reported	Not reported	Not reported	
Devlin et al. (2021)	Bitewing	24	CNNs	High	0.71	Not reported	Not reported	Not reported	Not reported	Not reported	Not reported	Not reported	
Bayrakdar et al. (2022)	Bitewing	53	CNNs	Not reported	0.84	Not reported	0.81	0.84	Not reported	Not reported	Not reported	Not reported	
Zheng et al. (2021)	Radiographs	127	CNNs	0.82	0.85	0.82	0.81	Not Reported	0.89	Not reported	Not reported	Not reported	
Lian et al. (2021)	Panoramic	89	CNNs	0.986	0.821	Not reported	Not reported	0.663	Not reported	Not reported	Not reported	Not reported	
Moran et al. (2021)	Bitewing	45	CNNs	0.733	Not reported	Not reported	Not reported	Not reported	Not reported	Not reported	Not reported	Not reported	
Mertens et al. (2021)	Bitewing	20	CNNs	Not reported	0.81	Not reported	Not reported	Not reported	0.89	Not reported	Not reported	Not reported	
Vinayahalingam et al. (2021)	Panoramic	100	CNNs	0.87	0.86	0.88		0.86	0.90	Not reported	Not reported	Not reported	
Lee et al. (2021)	Bitewing	50	CNNs	Not reported	0.6502	Not reported	0.6329	0.6414	Not reported	Not reported	Not reported	Not reported	
Hur et al. (2021)	Panoramic and CBCT	792	ANNs	Not reported	Not reported	Not reported	Not reported	Not reported	0.88 to 0.89	Not reported	Not reported	Not reported	
De Araujo Faria et al. (2021)	Panoramic	15	ANNs	0.988	Not reported	Not reported	Not reported	Not reported	0.9869	Not reported	Not reported	Not reported	
Mao et al. (2021)	Bitewing	83	CNNs	0.9030	Not reported	Not reported	Not reported	Not reported	Not reported	Not reported	Not reported	Not reported	
Bayraktar & Ayan, 2022	Bitewing	200	CNNs	0.9459	0.7226	0.9819	Not reported	Not reported	0.8719		0.8658	0.9564	
Zhu et al. (2022)	Panoramic	124	CNNs	0.9361	0.8601	Not reported	0.9409	0.9364	Not reported	Not reported	Not reported	Not reported	
Srivastava et al. (2017)	Bitewing	500	CNNs	Not reported	0.805	Not reported	0.615	0.70	Not reported	Not reported	Not reported	Not reported	
Yun et al. (2018)	Bitewing	40	CNNs	Not reported	Not reported	Not reported	0.546–0.418	0.697–0.584	Not reported	Not reported	Not reported	Not reported	
Khan et al. (2021)	Periapical	30	CNNs	Not reported	Not reported	Not reported	Not reported	0.239	Not reported	Not reported	Not reported	Not reported	
Jung & Kim (2020)	Periapical	25	CNNs	0.9847	Not reported	0.9953	Not reported	Not reported	Not reported	Not reported	Not reported	Not reported	
Geetha & Aprameya (2019)	Periapical	64	Support vector machine	0.968	Not reported	0.866	0.960	Not reported	Not reported	Not reported	Not reported	Not reported	

Risk of bias assessment

In this umbrella review, the quality assessment of the included studies used the Joanna Briggs Institute (JBI) checklist, comprising 11 criteria designed to evaluate various aspects of an SR.

The set of eleven questions (Q1–Q11) was carefully designed to target specific domains, serving as a reliable measure of study quality with high sensitivity and specificity. These questions were used to evaluate the quality of the study across eleven distinct domains (D1–D11).

The response options for each criterion are “Yes,” “No,” “Unclear,” or “Not Applicable.” Two reviewers independently evaluated each study included in the review. Disagreements were resolved through further discussions between the reviewers to achieve consensus. The overall quality of each study was determined by counting the number of criteria that received a “Yes” response. To categorize the quality of these studies, the total possible scores, ranging from 0 to 11, were divided into three levels: poor quality (0–3 scores), medium quality (4–7 scores), and high quality (8–11 scores), aligning with categorization methods used in other research contexts. It is important to note that the JBI checklist itself does not prescribe a specific methodology for assessing study quality.

Results

Screening of SRs and MAs

To perform a thorough literature search, we used the PubMed/Medline, Embase, Scopus, and Web of Science databases. Our initial search yielded 109 entries, which were carefully reviewed to remove duplicates. After this process, 55 studies remained for title and abstract review. Through detailed assessment, we identified 14 studies that met our inclusion criteria. After a thorough review of the full texts, we selected seven review articles to include in our comprehensive umbrella study (Prados-Privado et al., 2020; Khanagar et al., 2021; Musri et al., 2021; Khanagar et al., 2022; Mohammad-Rahimi et al., 2022; Singh & Raza, 2022; Talpur et al., 2022). Figure 1 shows a visual representation of the article selection process (adopted PRISMA flow diagram). Additionally, Table 2 lists the specific search terms used for each database and the corresponding number of results obtained from each.

Characteristics of the included SRs

One study that stood out for its comprehensive scope and high-quality reporting assessed by the JBI quality assessment checklist, was performed by Khanagar et al. (2022). The study investigated the effectiveness of AI-based models in identifying, diagnosing, and predicting DC. Out of 448 studies reviewed, 34 were included in the qualitative synthesis, using input datasets such as periapical radiographs, bitewing radiographs, panoramic radiographs, and other modalities to develop ANN and CNN models (Khanagar et al., 2022).

Despite differences between the included studies, all reviews used a similar approach and included various modalities (Prados-Privado et al., 2020; Khanagar et al., 2021; Musri et al., 2021; Mohammad-Rahimi et al., 2022; Singh & Raza, 2022; Talpur et al., 2022).

A notable point of contention across all included studies was the time limit for the inclusion of studies. While all reviews effectively screened and retrieved new articles based on their search periods, certain studies restricted their searches to articles indexed from specific years, such as 2010 (Mohammad-Rahimi et al., 2022), 2015 (Musri et al., 2021), and 2016 (Singh & Raza, 2022). The implications of these varying time limits warrant further discussion.

Regarding the overlapping included articles in these reviews, it should be noted that in accordance with the aim of this study, only studies that used “radiographic modalities input” were considered for the overlap assessment. The findings of this assessment are detailed in Table 5. Out of the 24 studies evaluated, only four were consistently included in multiple reviews. Specifically, the studies by Lee, Geetha, and Srivastav each appeared in four out of the seven reviews included in our current review. Meanwhile, the study performed by Choi, Eun & Kim (2016) was featured in three reviews.

Table 5 Overlapping of the articles between the SRs.

	Khanagar et al. (2021)	Khanagar et al. (2022)	Mohammad-Rahimi et al. (2022)	Musri et al. (2021)	Prados-Privado et al. (2020)	Singh & Raza (2022)	Talpur et al. (2022)	
Devito, de Souza Barbosa & Filho (2008)	Yes	No	No	No	Yes	No	Yes	
Lee et al. (2018)	Yes	Yes	No	No	Yes	Yes	Yes	
Choi, Eun & Kim (2018)	No	Yes	Yes	Yes	No	No	No	
Cantu et al. (2020)	No	Yes	No	No	No	No	No	
Geetha, Aprameya & Hinduja (2020)	No	Yes	No	No	Yes	Yes	Yes	
Chen et al. (2021)	No	Yes	No	No	No	No	No	
Devlin et al. (2021)	No	Yes	No	No	No	No	No	
Bayrakdar et al. (2022)	No	Yes	No	No	No	No	No	
Zheng et al. (2021)	No	Yes	No	No	No	No	No	
Lian et al. (2021)	No	Yes	No	No	No	No	No	
Moran et al. (2021)	No	Yes	No	No	No	No	No	
Mertens et al. (2021)	No	Yes	No	No	No	No	No	
Vinayahalingam et al. (2021)	No	Yes	No	No	No	No	No	
Lee et al. (2021)	No	Yes	No	No	No	No	No	
Hur et al. (2021)	No	Yes	No	No	No	No	No	
De Araujo Faria et al. (2021)	No	Yes	No	No	No	No	No	
Mao et al. (2021)	No	Yes	No	No	No	No	No	
Bayraktar & Ayan (2022)	No	Yes	No	No	No	No	No	
Zhu et al. (2022)	No	Yes	No	No	No	No	No	
Srivastava et al. (2017)	No	No	Yes	No	Yes	Yes	Yes	
Yun et al. (2018)	No	No	Yes	No	No	No	No	
Khan et al. (2021)	No	No	Yes	No	No	No	No	
Jung & Kim (2020)	No	No	Yes	No	No	No	No	
Geetha & Aprameya (2019)	No	No	No	No	No	Yes	No	

In each review, various quality assessment tools were used. Three SRs used the Quality Assessment of Diagnostic Accuracy Studies (QUADAS-2) tool guidelines (Khanagar et al., 2021, 2022; Mohammad-Rahimi et al., 2022), one followed Cochrane guidelines (Prados-Privado et al., 2020), and another used a self-created quality assessment checklist (Talpur et al., 2022). However, the quality assessment protocols and checklists used were inadequately described in the remaining two reviews (Musri et al., 2021; Singh & Raza, 2022).

Three of the included reviews adhered to the PRISMA–DTA guidelines as their reporting framework (Khanagar et al., 2021, 2022; Mohammad-Rahimi et al., 2022). Additionally, two other reviews followed the original PRISMA checklist for their reviews (Musri et al., 2021; Talpur et al., 2022). However, the remaining two reviews did not specify the guidelines or checklists used for their study reports (Prados-Privado et al., 2020; Singh & Raza, 2022).

Risk of bias and quality assessment of included studies

Three out of the seven included studies were classified as “high-quality” according to the JBI quality assessment checklist (Prados-Privado et al., 2020; Khanagar et al., 2022; Talpur et al., 2022). All remaining studies were categorized as “moderate-quality” (Figs. 2, 3) (Khanagar et al., 2021; Musri et al., 2021; Mohammad-Rahimi et al., 2022; Singh & Raza, 2022).

Figure 2 Evaluation of individual bias risk was assessed in the chosen systematic reviews.

Each study was evaluated for quality across eleven distinct domains (D1-D11) using the Joanna Briggs Institute (JBI) checklist for quality assessment.

Figure 3 An overall assessment of the risk of bias in the chosen systematic reviews comprises eleven carefully crafted questions (Q1–Q11) that target specific domains.

This assessment method, based on the Joanna Briggs Institute (JBI) checklist for quality appraisal, serves as a dependable measure of study quality with high sensitivity and specificity.

According to the JBI checklist, a main concern in SRs is the method of critical appraisal for included studies. To minimize bias and systematic errors in a systematic review, two or more reviewing team members must independently perform the critical evaluation of included studies, and this process should be duplicated. The review must clearly state that at least two reviewers independently performed the critical appraisal and consulted with each other as needed. This approach ensures that decisions regarding study quality and eligibility are based on a thorough and unbiased assessment (Aromataris et al., 2015). However, this issue is typically not comprehensively addressed in the text of several included reviews (Khanagar et al., 2021; Musri et al., 2021; Singh & Raza, 2022; Talpur et al., 2022).

A more critical issue in some included studies was the use of inadequate sources, which may have resulted in not covering all available evidence for the review (Musri et al., 2021; Singh & Raza, 2022). An SR needs to encompass all available evidence, which requires a thorough and comprehensive search strategy. This involves searching multiple electronic databases, including key bibliographic citation repositories. Additionally, a well-performed review should incorporate a search for grey literature or unpublished studies. This may involve exploring relevant websites or repositories of theses (Aromataris et al., 2015). Undertaking such extensive searches is crucial to minimize the risk of publication bias in an SR.

Evidence from high-quality reviews

In their 2022 study, Khanagar et al. (2022) aimed to assess the diagnostic accuracy and effectiveness of AI-based models in detecting, diagnosing, and predicting DC. After a thorough analysis, 34 articles meeting the selection criteria were critically reviewed. The study concluded that AI models demonstrate remarkable efficacy and are suitable for clinical application in identifying patients at higher risk of DC. These models also enhance diagnostic accuracy, treatment quality, and patient outcomes. The predictive capabilities of these models are valuable for developing preventive dental care strategies and personalized oral hygiene and dietary plans for individuals prone to DC. Despite their impressive performance, the study identified limitations in the size and diversity of the datasets used in the articles. Therefore, further training and validation of these models are necessary to enhance their effectiveness (Khanagar et al., 2022). It is important to note that this study encompassed various modalities for model development, including tabular, 2D, and 3D data. The conclusions drawn from different modalities may need modification when evaluating a certain modality, such as radiographs. Ultimately, eighteen out of the 34 studies included in Khanagar et al.’s (2022) review were selected for further evaluation in accordance with the aims of this overview.

In 2020, Prados-Privado et al. (2020) reviewed the latest advancements in neural networks for caries detection and diagnosis. This review encompassed thirteen studies using techniques such as near-infrared light transillumination and periapical and bitewing radiography. The studies included varied image databases, ranging from 87 to 3,000 images, with an average of 669 images per study. Seven studies involved expert dentists who annotated the presence of DC in each image. Not all studies provided a detailed definition of caries or specified the type of carious lesion detected. The review highlighted the variability in diagnostic efficacy among AI models using different algorithms, emphasizing the need for verification of their generalizability and reliability. Before these models can be effectively implemented in clinical practice, a comparative analysis of the results from each algorithm is necessary (Prados-Privado et al., 2020). Four studies from this review were selected for further evaluation in this overview.

In their 2022 study, Talpur et al. (2022) performed an SR to investigate the correlation between DC and ML. They identified 133 articles and eventually included twelve in their review. The authors evaluated the application of DL in dental imaging for diagnosing DC, specifically proximal, occlusal, and root caries. The SR concluded that these twelve studies provided substantial evidence in distinguishing dental X-ray images with and without caries. Among these algorithms, the neural network backpropagation algorithm was identified as the most effective, achieving up to 99% accuracy in detecting DC (Talpur et al., 2022). The study emphasizes the need for further extensive research to diagnose various types of DC, taking into account disease progression (chronic, acute, and arrested), which could offer insights into the severity and extent of caries. Additionally, four studies from this review were chosen for detailed analysis.

Evidence from moderate-quality reviews

In 2020, Khanagar et al. (2021) performed a study to evaluate the development and efficacy of AI applications in dentistry, focusing on their role in diagnosis, clinical decision-making, and prognosis prediction. Following specific inclusion and exclusion criteria, they selected 43 articles for review. The review found that most studies used AI models based on CNNs and artificial neural networks (ANNs). These models have demonstrated the ability to achieve or exceed the precision and accuracy of trained dental specialists. The authors suggested that AI could serve as a complementary tool for dentists, enhancing diagnostic accuracy, assisting in treatment planning, and improving outcome predictions. Furthermore, AI and DL systems could provide diagnostic support to general dentists, thereby saving time and increasing efficiency. The use of AI to provide second opinions and improve diagnostic accuracy was also highlighted (Khanagar et al., 2021).

Mohammad-Rahimi et al. (2022) performed an SR to evaluate DL studies focused on caries detection. They included studies that used DL models with various dental imaging modalities, such as radiographs, photographs, and optical coherence tomography images, published after 2010. Out of 252 potential references, 48 studies were fully reviewed, with 42 ultimately included in their analysis. The accuracy of caries detection models varied, achieving 82–99.2% on periapical radiographs, 87.6–95.4% on bitewing radiographs, and 86.1–96.1% on panoramic radiographs. While the accuracy was promising, they noted that the overall quality and reporting of the studies were generally low. They concluded that DL models could aid in decision-making regarding the presence or absence of caries (Mohammad-Rahimi et al., 2022). Five studies from this review were selected for further evaluation in this overview.

Musri et al. (2021) focused on evaluating DL and CNNs for detecting and diagnosing early-stage DC using periapical radiographs. Their research, encompassing evidence from 2015 to 2021, revealed that deep CNN-based detectors effectively identified caries, enabling practitioners to discern changes in the location and morphology of DC lesions. Despite generally small datasets used in the studies reviewed, both traditional and optimized DL methods were used, although the studies did not differentiate between early-stage and root caries. The authors noted that DL algorithms, with their complex and evolving layers, continue to advance, significantly enhancing their accuracy in object detection and segmentation. They concluded that CNN algorithms provide thorough, reliable, and accurate assessments of images and disease detection, thereby improving the efficiency and effectiveness of diagnosis and prognosis of DC in periapical radiographs (Musri et al., 2021).

Singh & Raza (2022) performed a study to explore recent advancements in DL-based analysis of dental and maxillofacial images, reviewing literature from 2016 to 2020. They included 75 articles that applied DL to dental image classification, segmentation, object detection, and other image-processing activities. The authors observed that AI and neural network applications in dental image analysis have demonstrated superior performance, occasionally surpassing that of dental experts. However, they highlighted the importance of validating the accuracy of these systems across various imaging devices and modalities to standardize data. Given the prevalence of conditions such as periodontally compromised teeth, tooth caries, and apical lesions, they suggested that compiling large datasets of similar conditions could help achieve higher levels of accuracy more efficiently (Singh & Raza, 2022).

Discussion

DC, a prevalent oral disease affecting individuals worldwide of all ages, is typically diagnosed through visual examination of dental surfaces. Unfortunately, this approach can be subjective and prone to errors. Radiographic imaging has become a widely adopted method for non-invasive and reliable detection of DC. However, this method can also pose challenges, especially in detecting small carious lesions or those in hard-to-reach areas. Moreover, the diverse methods and devices used to acquire radiographs can introduce bias in clinical decision-making. Early detection of DC can considerably decrease the necessity for complex treatments, thereby reducing overall treatment costs. Therefore, it is essential for clinicians to accurately and reliably identify DC through dental radiographs to ensure optimal care for their patients. This approach can increase the overall efficiency of the healthcare system.

The detection and prediction of DC through radiographic images have demonstrated considerable potential using DL and neural network algorithms. These algorithms employ ANNs trained on extensive datasets of radiographic images showing both healthy and decayed teeth. Once trained, the algorithms can accurately identify and predict the presence of carious lesions in new radiographic images.

The application of AI in detecting DC has produced impressive results, as demonstrated by Lee et al.’s (2018) study. This study highlighted the effective use of CNN algorithms to identify and diagnose DC through periapical radiographs, achieving significant performance levels. Similarly, Devito, de Souza Barbosa & Filho (2008) implemented an AI-powered ANN model for diagnosing proximal caries using bitewing radiographs, yielding promising results. Additionally, Cantu et al. (2020) developed a DL model for DC detection on bitewing radiographs, surpassing the accuracy of skilled dentists in identifying early-stage lesions.

Various studies were performed to address this issue and evaluate the technology (Prados-Privado et al., 2020; Khanagar et al., 2021; Musri et al., 2021; Khanagar et al., 2022; Mohammad-Rahimi et al., 2022; Singh & Raza, 2022; Talpur et al., 2022). It is important to note that each original study may introduce new AI models for detecting and predicting DC. As this field progresses, efforts have been made to systematically review the existing evidence to provide a valuable resource for clinical decision-making.

Prados-Privado et al. (2020) analyzed existing research and found that while studies on DC detection report similar metrics, direct comparisons between them are not feasible. This is attributed to each AI system being specifically designed for particular tasks, making cross-study comparisons impractical. The authors recommend that future reviews should focus on studies using AI for similar purposes (Prados-Privado et al., 2020).

The field of DL in caries detection is rapidly advancing, with noticeable differences in approaches and outcomes across various studies. Multiple SRs and MAs have been performed to evaluate the accuracy, sensitivity, and specificity of DL algorithms in detecting DC. This comprehensive umbrella review aims to assess published SRs and MAs that examine the effectiveness of DL and neural network algorithms in detecting and predicting DC from radiographic images. The goal is to consolidate and analyze the findings from these studies.

In this overview, 77 records were initially identified through electronic database searching, from which seven SRs met the eligibility criteria and were selected for further evaluation. Despite differences in their specific aims and focus areas, the relevant studies included in each SR, according to the objectives of this overview, were retrieved for further evaluation.

The quality assessments of these seven SRs reveal that three are classified as “high-quality” (Prados-Privado et al., 2020; Khanagar et al., 2022; Talpur et al., 2022), while the remaining four are categorized as “moderate-quality” (Khanagar et al., 2021; Musri et al., 2021; Mohammad-Rahimi et al., 2022; Singh & Raza, 2022). Despite this assessment, it is noteworthy that some of these SRs lack critical features that could introduce bias into their reports and potentially affect the generalizability of the data.

Some studies limited their search to articles indexed from 2010 (Mohammad-Rahimi et al., 2022); 2015 (Musri et al., 2021), and 2016 (Singh & Raza, 2022). In contrast, the “high-quality” SRs did not impose such restrictions and included relevant studies from 2008 (Prados-Privado et al., 2020, Khanagar et al., 2022).

The importance of this issue becomes apparent when examining one of the first studies on the development of an AI model for DC detection. In 2008, Devito, de Souza Barbosa & Filho (2008) performed research to assess whether using a multilayer perceptron neural network, an AI model, could enhance the diagnosis of proximal caries on radiographs. Their findings revealed that the AUC for the best of the 25 examiners was 0.717, whereas the neural network achieved an AUC of 0.884, indicating a substantial improvement in diagnosing proximal caries. This indicates that considering all examiners, the neural network achieved a diagnostic improvement of 39.4% (Devito, de Souza Barbosa & Filho, 2008).

Furthermore, some of the SRs lacked quality assessments of the included studies, which is a considerable source of bias that could potentially affect the reliability of their results and conclusions (Musri et al., 2021; Singh & Raza, 2022).

It should be emphasized that this study specifically focused on reviews that included radiographic modalities when assessing overlapping articles. The results of this assessment are outlined in Table 5. Out of the 24 studies evaluated, only four were found to be included in multiple reviews. Specifically, the studies by Lee et al. (2018), Geetha, Aprameya & Hinduja (2020), and Srivastava et al. (2017) were each included in four out of the seven reviews, while the study by Choi, Eun & Kim (2016) was featured three times.

Lee et al. (2018) used a deep CNN model based on U-Net to detect DC in bitewing radiographs. This model achieved an accuracy of 63.29%, a recall of 65.02%, and an F1-score of 64.14%. However, a limitation of this study was the relatively small dataset used for developing the model.

In another study, Geetha, Aprameya & Hinduja (2020) explored an AI-based model for DC diagnosis in digital radiographs. This model demonstrated exceptional performance, achieving an accuracy rate of 97.1%, a false positive rate of 2.8%, and a receiver operating characteristic (ROC) area of 0.987. Despite these promising results, further refinement is necessary for the accurate classification of DC based on lesion depth.

Srivastava et al. (2017) used a substantial annotated dataset comprising more than 3,000 bitewing radiographs to develop an automated system for diagnosing DC. Their approach integrated a deep, fully convolutional neural network (FCNN) with over 100 layers, specifically trained for caries detection in bitewing radiographs. When compared to the diagnostic capabilities of three certified dentists, their system exhibited superior performance in terms of recall (sensitivity) and F1-score (accuracy of detection).

Choi, Eun & Kim (2016) discovered that their system, which used a CNN combined with crown extraction in various periapical images, outperformed a system using a standard CNN in their tests.

Khanagar et al. (2022) reported that the DL model CapsNet effectively processes visual elements such as posture, speed, hue, and texture, thereby enhancing its capabilities for detecting and diagnosing DC (He et al., 2016). Despite these advancements, the study acknowledged limitations, including the omission of certain clinical factors, a limited number of radiographs used, and a focus solely on permanent teeth (Sabour, Frosst & Hinton, 2017; Lee et al., 2021).

Cantu et al. (2020) developed a DL model for DC detection in bitewing radiographs, achieving impressive accuracy, sensitivity, and specificity values of 0.80, 0.75, and 0.83, respectively. Notably, this model outperformed experienced and trained dentists in identifying initial caries lesions. A considerable strength of this study was the use of a large, balanced dataset for both training and testing.

Talpur et al. (2022) noted that the studies they reviewed collectively endorsed the application of DL and deep neural networks in detecting DC. These studies used various ML approaches, including backpropagation, FCNN, CNN, mask CNN, support vector machine (SVM), and ANN. Patil, Kulkarni & Bhise (2019) performed a comparative study of different algorithms for diagnosing tooth caries, specifically focusing on SVM, adaptive dragonfly algorithm neural network (ADA–NN), naïve Bayes (NB), and k-nearest neighbor (KNN). Their analysis involved 120 dental images divided into three sets of 40 images each. The ADA–NN emerged as the most effective, surpassing the other algorithms by 5.5%, 11.76%, and 6.5%. Furthermore, Devito, de Souza Barbosa & Filho (2008) and Geetha, Aprameya & Hinduja (2020) both used the backpropagation DL algorithm for predicting DC. Devito, de Souza Barbosa & Filho (2008) specifically focused on predicting proximal DC from X-rays, achieving an accuracy of 88.4%.

DL algorithms have also demonstrated potential in detecting and predicting DC using radiographic images. De Araujo Faria et al. (2021) performed research on an AI model designed for predicting and detecting radiation-related caries (RRC) in panoramic radiographs. This model achieved impressive accuracy levels, with a detection accuracy of 98.8% and an AUC of 0.9869. For prediction, it demonstrated an accuracy of 99.2% and an AUC of 0.9886. However, Prados-Privado et al. (2020) expressed concerns regarding the small sample size used in this study. They noted that the patients at the specific center involved were frequently in advanced stages of DC during their radiographic imaging (Khanagar et al., 2022). In a related context, Hung et al. (2019) acknowledged the impressive capabilities of AI technology in accurately predicting root caries.

One advantage of DL algorithms is their ability to identify carious lesions that human examiners may overlook. Schwendicke, Tzschoppe & Paris (2015) highlighted that numerous studies have consistently shown the high accuracy of DL in caries detection, often achieving sensitivity and specificity rates exceeding 80%. Importantly, in several of these studies where human experts served as benchmarks (not just reference standards), DL demonstrated equal or superior accuracy compared to human judgment. Given the typically moderate accuracy of dentists, particularly in early lesion detection, DL appears to provide considerable accuracy for this purpose.

Devlin et al. (2021) performed a study aimed at developing an AI model capable of detecting enamel-only proximal DC using bitewing radiographs. The results of this model were promising, demonstrating performance levels comparable to those of expert dentists. In a separate study, Bayraktar & Ayan (2022) explored AI-driven DL models, namely VGG-16 and U-Net, for the automated detection and segmentation of caries in bitewing radiographs. These models outperformed experienced dental specialists despite limitations imposed by a small dataset from a single center. Zheng et al. (2021) compared three CNN models—VGG, Inception V3, and ResNet. The ResNet model exhibited superior performance among the three, surpassing even the capabilities of trained dentists. However, it is important to note that the diagnostic decisions in this study were made by a panel of experienced dentists, which is not considered the optimal benchmark for diagnosing deep caries and pulpitis.

While DL algorithms show promising results, their effectiveness depends on access to large datasets of high-quality radiographic images for training. The accuracy of these algorithms can vary based on the type and quality of the radiographs and the expertise of the users. Therefore, continuous assessment and refinement of these algorithms are essential to enhance their accuracy and reliability.

Mohammad-Rahimi et al. (2022) highlighted a notable deficiency in the reviewed studies: insufficient detail on how annotator validation and calibration were managed, especially regarding resolving disagreements between annotators. Future studies should focus on improving reporting standards by adhering to guidelines such as STARD–AI for AI-based diagnostic studies, CLAIM for AI in medical imaging, and recent guidelines specific to dental AI (Mongan, Moy & Kahn, 2020; Sounderajah et al., 2020; Schwendicke et al., 2021).

Prados-Privado et al. (2020) noted a lack of information on inter- and intra-examiner consistency in the reviewed studies. Typically, Cohen’s kappa is used to assess these agreements. Techniques such as ROC curves or Bland–Altman plots could help identify samples with less consensus. It is important to note that manual labeling by experts while serving as a reference for training and evaluation, may not represent absolute truth. Therefore, a histologic gold standard method is essential to validate caries diagnosis methods. Unfortunately, none of the reviewed studies specified the reference standards used.

Mertens et al. (2021) investigated a CNN model for detecting proximal DC using bitewing radiographs, achieving a ROC of 0.89 and a sensitivity of 0.81. This performance was superior compared to evaluations by five expert dentists. Khanagar et al. (2022) commended the study’s methodology, emphasizing its randomized controlled trial design. However, a major limitation of the study was its reliance on a limited dataset obtained from a single center.

An accurate definition of DC and the specific type of carious lesions being investigated is essential for effectively comparing and interpreting results across studies. In Prados-Privado et al.’s (2020) review, studies using the ICDAS II reported accuracies ranging from 80% to 88.9%. Conversely, a study that defined caries as a radiolucent mineralization loss achieved an accuracy of 97.1%. However, most of the studies reviewed did not provide a detailed definition of caries lesions.

Prados-Privado et al. (2020) highlighted a significant bias factor concerning the training dataset. They noted that experts should annotate the images used for training. However, only half of the studies they included indicated the involvement of examiners in image labeling, with variations in the experience and number of these examiners across different studies (Prados-Privado et al., 2020).

Bussaneli et al. (2015) found in their research that while the examiner’s experience did not significantly affect the detection of occlusal lesions in primary teeth, it did influence treatment decisions for initial lesions. However, when AI is trained using evaluations from human observers, its performance is inherently limited by the quality of the input. Overfitting is a critical issue in AI technology. According to Anderson & Burnham (2004), overfitting occurs when random variation is mistakenly interpreted as part of the model’s structure. A model becomes overfitted when it is overly adjusted to fit the original data, thereby compromising its application to new data and resulting in suboptimal results (Mutasa, Sun & Ha, 2020).

Mohammad-Rahimi et al. (2022) emphasized the necessity for future studies to identify optimal strategies for establishing a reference test that accurately reflects the “truth,” including methods such as majority voting, stepwise annotation, or triangulation with secondary imaging. Furthermore, these studies should integrate thorough and accurate data labeling processes to develop robust AI models (Willemink et al., 2020).

Limitations

The current models exhibit strong performance, but they are constrained by limitations related to the size and diversity of the datasets used in the studies. Consequently, further training and validation are essential to optimize their performance. This study provides a thorough and systematic evaluation of DL in caries detection, facilitating a thorough synthesis and comparison of the results. It is important to note that our search was limited to a specific time period and scope, focusing solely on DL while excluding alternative image analysis methods previously used. These constraints were justified by the substantial and diverse evidence we obtained. Despite our intention to conduct an MA, the extensive variability and inadequate reporting quality in the studies limited our ability to gather the necessary details for such an analysis.

Strengths

This is the first overview in this field aimed at identifying knowledge gaps and evaluating the quality of SR studies performed in this context. Our findings suggest that CNN algorithms provide thorough, reliable, and accurate image analysis and disease detection. CNN algorithms facilitate effective and efficient diagnosis and improving the prognosis of DC in periapical radiographs.

Future directions

The studies referenced in the SRs used relatively small datasets for traditional and advanced DL methods without distinguishing between early and root caries. There is a need for accurate detection and diagnosis of DC to reduce the costs associated with oral health management and enhance the likelihood of preserving natural teeth. Many researchers advocate for the development of a large-scale, publicly accessible dataset to achieve clinically reliable accuracy. It is also important to focus on automating preprocessing tasks using unsupervised or semi-supervised training to reduce errors from manual procedures.

Despite the promising advancements in the application of deep learning (DL) and convolutional neural networks (CNN) for detecting dental caries (DC) using radiographic images, several limitations persist that require attention from future research efforts.

Weaknesses in existing methods

Dataset size and diversity: Many current studies rely on relatively small and homogeneous datasets, which limits the generalizability and robustness of the models. These datasets often lack diversity in terms of demographic variables, imaging modalities, and the extent of disease progression. Consequently, the models trained on such datasets may perform well in controlled environments but struggle to maintain accuracy when applied to broader, real-world scenarios.

Overfitting issues: The risk of overfitting remains a significant concern, particularly when models are trained on small datasets with insufficient variability. Overfitting occurs when a model learns to perform well on the training data but fails to generalize to new, unseen data. This compromises the model’s applicability in clinical settings.

Lack of standardization: There is a notable lack of standardization in the training and validation processes across different studies. Variations in the types of radiographic images, annotation protocols, and quality assessment methods hinder the ability to compare results directly and derive consistent conclusions.

Insufficient attention to clinical relevance: While many DL models achieve high accuracy rates, there is often insufficient focus on ensuring that these models are clinically relevant and actionable. Models may detect lesions with high sensitivity, but without considering the clinical decision-making process, such detections may not translate into improved patient outcomes.

New directions for future research

Development of large-scale, diverse datasets: Future research should prioritize the creation and utilization of large, diverse, and publicly accessible datasets. These datasets should encompass a wide range of patient demographics, imaging modalities, and varying degrees of caries severity. This will improve the generalizability and robustness of AI models, making them more reliable in diverse clinical settings.

Focus on unsupervised and semi-supervised learning: To reduce the reliance on large labeled datasets and minimize manual errors, researchers should explore unsupervised or semi-supervised learning techniques. These approaches can leverage unlabeled data, which is abundant, to improve model training and performance without extensive manual annotation.

Integration of multi-modal data: Combining data from various imaging techniques (e.g., periapical radiographs, bitewing radiographs, optical coherence tomography) could enhance the accuracy of DL models. Future studies should investigate how integrating multi-modal data can improve diagnostic accuracy and provide a more comprehensive assessment of dental caries.

Improvement in interpretability and explainability: As AI models become more complex, it is crucial to develop methods that improve the interpretability and explainability of these models. Clinicians need to understand how AI algorithms arrive at their conclusions to trust and effectively use these tools in practice. Research should focus on creating models that provide clear, understandable outputs that align with clinical reasoning.

Personalized dental care algorithms: Future work should aim to develop AI models that can contribute to personalized dental care. This involves creating algorithms that not only detect caries but also predict disease progression, suggest tailored treatment plans, and provide patient-specific preventive measures based on individual risk factors.

Validation and benchmarking across multiple sites: It is essential to validate AI models across multiple clinical settings and institutions to ensure their generalizability and reliability. Future research should focus on cross-institutional studies that benchmark the performance of DL algorithms, using standardized protocols and evaluation metrics.

Exploration of new algorithms and architectures: Continued exploration of novel DL architectures, such as transformers and attention-based models, may yield further improvements in caries detection and diagnosis. These advanced architectures could offer better performance in handling complex patterns in medical imaging data, providing more accurate and reliable diagnostics.

Conclusions

1) The reviewed studies consistently demonstrate that DL and CNN models can achieve high levels of accuracy in identifying and predicting DC. These models often match or surpass the diagnostic capabilities of trained dental professionals, highlighting their potential as powerful tools in dental care.

2) A recurrent limitation across the studies is the reliance on small and homogeneous datasets. This restricts the models’ ability to generalize across diverse populations and varying clinical conditions, underscoring the need for larger, more diverse datasets in future research.

3) The risk of overfitting and the lack of standardization in training, validation, and quality assessment methodologies pose significant barriers to the broader clinical application of these AI models. Addressing these issues is crucial for the development of robust, reliable AI tools.

4) While DL models show high technical performance, there is a pressing need to ensure that these models are clinically relevant and interpretable. Models must be designed and tested with real-world clinical decision-making processes in mind to be truly effective in improving patient care.

5) Further comprehensive studies are required to assess the practicality and effectiveness of these algorithms in everyday dental settings.

Supplemental Information

Supplemental Information 1 PRISMA 2020 checklist.

Supplemental Information 2 Extracted PubMed database.

Supplemental Information 3 Extracted Scopus database.

Supplemental Information 4 Extracted Scopus secondary database.

Supplemental Information 5 Extracted Web of Science database.

Supplemental Information 6 Extracted Embase database.

Supplemental Information 7 Systematic Review and or Meta-Analysis Rationale.

Supplemental Information 8 The Joanna Briggs Institute (JBI) checklist questions for quality assessments of systematic reviews.

Additional Information and Declarations

Competing Interests

Author Contributions

Data Availability

The authors declare that they have no competing interests.

Mahmood Dashti conceived and designed the experiments, authored or reviewed drafts of the article, and approved the final draft.

Jimmy Londono conceived and designed the experiments, authored or reviewed drafts of the article, and approved the final draft.

Shohreh Ghasemi performed the experiments, authored or reviewed drafts of the article, and approved the final draft.

Niusha Zare performed the experiments, authored or reviewed drafts of the article, and approved the final draft.

Meyassara Samman performed the experiments, authored or reviewed drafts of the article, and approved the final draft.

Heba Ashi performed the experiments, authored or reviewed drafts of the article, and approved the final draft.

Mohammad Hosein Amirzade-Iranaq analyzed the data, prepared figures and/or tables, authored or reviewed drafts of the article, and approved the final draft.

Farshad Khosraviani analyzed the data, prepared figures and/or tables, authored or reviewed drafts of the article, and approved the final draft.

Mohammad Sabeti analyzed the data, prepared figures and/or tables, authored or reviewed drafts of the article, and approved the final draft.

Zohaib Khurshid conceived and designed the experiments, authored or reviewed drafts of the article, and approved the final draft.

The following information was supplied regarding data availability:

This is a literature review.

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
