# Peer review of "Comparative analysis of deep learning algorithms for dental caries detection and prediction from radiographic images: a comprehensive umbrella review"

_PeerJ Computer Science, doi:10.7717/peerj-cs.2371_

## Round 0.1 · original submission · Major Revisions

We have completed the review process. Reviewers have suggested a major revision. You are required to address all the comments and suggestions, and submit a revision. In case review suggested some manuscript to cite, it is up to you to decide whether it is relevant to cite or not.

**Language Note:** The review process has identified that the English language must be improved. PeerJ can provide language editing services - please contact us at [email protected] for pricing (be sure to provide your manuscript number and title). Alternatively, you should make your own arrangements to improve the language quality and provide details in your response letter. – PeerJ Staff

Reviewer 1 ·

Basic reporting

Authors have conducted a review on evaluation of accuracy of deep learning and neural networks algorithms in detection and prediction of dental caries by
using radiographical images
In the very begining I must say that English language needs thorough revision . Look at the following senetnce from abstract only "The reported systematic reviews used a total 24 articles were used in those systematic reviews which used 2D radiographical images for caries detection".

Experimental design

There are lots of authors listed in this manuscript but the content seems very poor .Look at english used in this sentence: "After using
this tool/service, all of the authors reviewed and edited the content as needed and take full responsibility for the content of the publication."

A comparative study of the literature surveyed needs to put in a tabular from along with limitations, strengths, the dataset used, the algorithm used and accuracy achieved.

Then a seperate section needs to be added highlighting the overall summary of the results achieved by the authors [in the literature review]

Future directions section needs to be added which should highlight what weaknesses are there in the existing methods, what new directions,areas, and algorithms should be focussed on by future researchers in order to have an improvement in this area. This has to be again a separate section.

Carrying out the literature review is a huge responsibility and authors of this paper need to contribute heavily if they have taken the review process.

Validity of the findings

It is advised that authors either hire an English expert or use some premium services to improve the language. It is very basic.

Authors should focus on the review. I don't understand why more focus has been given to "Intended Audience" section. It can be written in just 3 to 4 lines.

Introduction is very very short. It should be extended to almost 3 pages.

Above comments need to be addressed one by one.

Reviewer 2 ·

Basic reporting

The proposed manuscript “Evaluation of accuracy of deep learning and neural networks algorithms in detection and prediction of dental caries by using radiographical images: An umbrella review” contains novel elements. However, it presents some aspects that need to be solved before reconsideration.
It is recommended to make some taxonomy figures for these related works to assist readers in understanding your work.

Experimental design

1. Authors should explain the duration of the timeline from articles considered in this survey under the Inclusion/Exclusion criteria.
2. The evaluation metrics for dental caries classification, detection, and segmentation, should be given in the literature.
3. The author ensures to follow the guidelines provided while reviewing the AI-based medical image analysis task.
Joseph R. England & Phillip M. Cheng, (2018). Artificial Intelligence for Medical Image Analysis: A Guide for Authors and Reviewers, American Journal of Roentgenology. https://doi.org/10.2214/AJR.18.20490
4. Please give a more detailed description of the potential drawbacks or limitations of dental caries detection using deep CNNs architecture under the discussion section, e.g., overfitting or domain shift.
5. In the conclusion section, the discussion of future direction is insufficient. Please give a more detailed description of the future research directions to promote the effect of deep-learning-based diagnosis models for dental caries in dental radiographs. I think it is very important to provide some research opportunities for the readers.
6. Table 4, has been compiled without any specific planning, to obtain a clear understanding author should rearrange Table 4.

Validity of the findings

NA

Additional comments

NA

---

## Round 0.2 · Major Revisions

Reviewers have raised concern regarding some of the previous comments. You are required to address all the comments and suggestions of reviewers (not addressed in the previous revision). I also have some concerns:

1) Title is too lengthy, and unclear. The two terms in title "deep learning" and "neural networks" somehow refer to similar entities in the manuscript. There are other terms as well that can be dropped. A suggested title is "Comparative analysis of deep learning algorithms for dental caries detection and prediction from radiographic Images: A comprehensive umbrella review"
2) Table 3, row 6, "Singh, N. K., Reza. Kh. (2022) (15)", "Articles that are yet to be published (in press)" need to be corrected. This article is already published. Update its volume and page number in the reference.
3) Some of the important papers on deep learning based architecture for dental caries detection are missed. If authors wish, they may include the followings: https://doi.org/10.1007/978-3-031-35501-1_3, https://doi.org/10.1007/978-981-99-1648-1_19. However, it is not mandatory. Its inclusion is solely up to the wishes of the authors.
4) Line 17-172, the articles published up to the end of August 2023 have been considered. Many of the new papers published during 2023–2024 have been missed. I strongly recommend authors include literature up to May 2024.
5) Table 5. presents "Overlapping of the articles between the SRs". What does this table contribute to?
6) Fig. 2, the description of columns D1, D2, ... D11 needs to be described in the figure legend.
7) Fig. 3, the description of the abbreviations Q1, Q2, ... Q11 needs to be described in the figure legend.

Reviewer 1 ·

Basic reporting

All of my comments have not been addressed properly. Authors need to address all the points.


limitations, strengths in the comparative study
Then a seperate section needs to be added highlighting the overall summary of the results achieved by the authors [in the literature review.

The Future Directions mentioned in the paper is nothing at all. please remove that added one. Address the comment properly as "Future directions section needs to be added which should highlight what weaknesses are there in the existing methods, what new directions,areas, and algorithms should be focussed on by future researchers in order to have an improvement in this area. This has to be again a separate section."

Authors should focus on the review. I don't understand why more focus has been given to "Intended Audience" section. It can be written in just 3 to 4 lines. (This does not substantiate it)

It is advised that authors either hire an English expert or use some premium services to improve the language.


Authors need to address above points in letter and sprit

Experimental design

no comment

Validity of the findings

no comment

Reviewer 2 ·

Basic reporting

Regarding the previous comment under basic reporting, the authors should update the pending taxonomy figure.

Experimental design

1. My previous comment "Authors should explain the duration of the timeline from articles considered in this survey under the Inclusion/Exclusion criteria" is not incorporated in the present form of the manuscript.
2. The authors failed to produce any substantial drawbacks and limitations of analyzing dental caries detection using deep CNN architecture under the discussion section.

Validity of the findings

Authors should address the comments carefully with significant literature.

Additional comments

NA

---

## Round 0.3 · Major Revisions

One of the reviewers raised a concern that all the suggested comments were not addressed in the revised manuscript. You are required to address those comments and resubmit.

Reviewer 1 ·

Basic reporting

I had given the comment as ""Future directions section needs to be added which should highlight what weaknesses are there in the existing methods, what new directions,areas, and algorithms should be focussed on by future researchers in order to have an improvement in this area. This has to be again a separate section." It is still not addressed as per the suggestions given

Why to keep the Intended Audience section which does not convey any sense?.

Journal guidelines have not been followed for citations. Authors should know and read Peerj author guidelines before submitting. What type of citation style does Peerj follow? is it APA,IEEE etc. It is as simple as that. Authors need to look into it properly.

Authors have produced the english editing certificate. I wonder there are lots of mistakes
Look at this "Our findings suggest that DL CNN algorithms
513 provide thorough, reliable, and accurate image analysis and disease detection, thereby facilitating
514 effective and efficient diagnosis and improving the prognosis of DC in periapical radiographs.
"
There are so many ands used and it seems very lengthy sentence.
Can't we write "DL CNN algorithms" above as Our findings suggest that DL based algorithms such as CNN..
Conclusions is too short and it does not wrap the manuscript. It needs to be comprehensive.

Authors need to improve the manuscript a lot

Experimental design

no comment

Validity of the findings

no comment

Additional comments

no comment

---

## Round 0.4 · accepted · Accept

We have completed the peer review process. I am happy to accept your manuscript for the publication in the current form.

Reviewer 1 ·

Basic reporting

seems ok now

Experimental design

no comment

Validity of the findings

no comment